# Implications of Temperature Abuse on Unpasteurized Beer Quality Using Organoleptic and Chemical Analyses

**DOI:** 10.3390/foods9081032

**Published:** 2020-08-01

**Authors:** Zahra H. Mohammad, Christopher C. Ray, Jack A. Neal, Glenn Cordua, Aaron Corsi, Sujata A. Sirsat

**Affiliations:** Conrad N. Hilton College of Hotel and Restaurant Management, University of Houston, Houston, TX 77204-3028, USA; zhmohamm@Central.UH.EDU (Z.H.M.); c.ray19@gmail.com (C.C.R.); JNeal@Central.UH.EDU (J.A.N.); GDCordua@Central.UH.EDU (G.C.); ajcorsi@Central.UH.EDU (A.C.)

**Keywords:** unpasteurized beer, sensory evaluation, chemical breakdown, temperature, beer quality

## Abstract

Beer flavor and sensory quality are affected by storage time and temperature due to chemical breakdown and aging. This study aimed to investigate the organoleptic properties of temperature-abused, unpasteurized craft beer and analyze the chemical breakdown associated with the process. Sensory tests were performed using a triangle test to determine consumer identification of temperature-abused beer. The chemical tests were conducted to determine the chemical breakdown of the two beer groups: control beer (COB) and temperature-abused beer (TAB). The chemical analysis of the two beer groups showed significant changes in multiple chemical compounds such as ethyl esters, linear aldehydes, and sulphur-compounds; however, the sensory analysis results were not significant even though 39% of participants were able to detect differences. in this study, two factors identified that caused chemical reactions in the TABs were oxidation and live yeast cells. In conclusion, these results can be used by beer producers to ensure a quality product throughout the distribution chain by controlling time and temperature.

## 1. Introduction

Beer is one of the most popular alcoholic beverages and is consumed in almost every country in the world in vast quantities [1,2]. Beer is a natural product that deteriorates at different rates depending upon the rate of time and temperature exposure [3]. The storage temperature of beer at various stages of the delivery chain affects chemical reactions that control the aging characteristics of the beer [3,4]. From brewery to consumer, commercial beer is exposed to several temperature fluctuations that can alter sensory properties and cause negative aging effects due to the chemical reactions in the beer over time [5,6]. According to Čejka et al. [3], during beer storage, the following processes occur: formation of stale flavors, haze and browning, reduction in bitterness, and decrease in fruity aroma. Thus, beer quality gradually decreases with elapsed time in storage.

Consumers assess several characteristics of beer to determine the overall quality [7] including sensory or taste [8,9]. Codină et al. [9] conducted a sensory analysis on Romanian beers and found a significant correlation between the sensory attributes and the chemical parameters through physico-chemical parameters that are responsible for the sensorial attributes of beer. Taste and aroma may be perceived separately, but they are often integrated to produce a total flavor impression [10]. Aroma is an important factor that the consumer perceives for assessing the quality of beer [7]. Beer aroma mostly consists of higher alcohols and esters [11]. A change in the concentration of these compounds may destroy the balance of the beer aroma [12]. Aroma control is a process where brewers control the possible conditions that can alter the beers aromatic properties during the brewing process. Researchers have isolated the various essential volatile compounds that affect beer flavor [13]. The chemical classes associated with these compounds include linear aldehydes, strecker aldehydes, ketones, cyclic acetyls, heterocyclic compounds, ethyl esters, lactones, and s-compounds. These chemical classes contain the most common chemical compounds found in beer that affects quality and flavor [13]. However, the volatile compounds profile of the beer is influenced by the yeast strain used for beer fermentation [14].

Marconi et al. [14] conducted two different experimental conditions where beer samples were either stored at 20 or 30 °C to investigate the influence of yeast strain, priming solution, and temperature on beer quality. The researchers also investigated the standard quality attributes, volatile compounds, and sensory profile of the bottle-conditioned beer. The authors found that the volatile profile was affected by the strain of yeast due to varied metabolic activity for each yeast strain. The authors also found that the Safbrew S-33^®^ yeast strain, when primed with Siromix^®^ and refermented at 30 °C, yielded the fastest formation of higher alcohols while maintaining low production of off-flavors. Finally, the authors concluded that a formulation of two yeast strains might reduce the time needed for bottle conditioning without affecting the quality of the final beer and improve efficiency and economic profits [14].

The oxidation of bitter beer acids and oxidation and polymerization of malt and hop polyphenols could also occur during beer storage [15]. These unwanted reactions also cause adverse effects on TAB during storage conditions. Oxygen can affect beer during storage by contributing to its deterioration and shortening the shelf life [12]. Oxygen deteriorates final packaged beer by altering appearance, color, taste, aroma, and most importantly, beer flavor [12]. In modern bottle filling equipment in the breweries is designed to achieve very low levels (0.03 mg/L) of O2 concentrations. Even with the low-level oxygen content of the final package beer bottles, there is still enough oxygen to formulate staling substances [16].

Few academic studies have been conducted to determine the effect of time-temperature abuse and agitation on the quality of beer. The primary focus of the previous literature to date regarding beer quality has been to determine the chemical deterioration that occurs in general and substandard storage conditions. In addition, the majority of craft beers are non-pasteurized. Currently, the predominant quality problem of beer can be attributed to the change in its chemical composition during storage, which may alter the sensory properties [5,13]. The overall objective of this study was to determine the effect of temperature abuse on the chemical breakdown, sensory properties of unpasteurized bottled beer, and the implications for the industry and manufactures.

## 2. Materials and Methods

The approach of this study is based on quasi-experimental design. According to Shadish and Galindo [17], a quasi-experimental design manipulates treatments to determine their effects. This design approach differs from randomized experiments in that units are not randomly assigned to conditions. The quasi-experimental design combined the multiple measurements of the chemical analysis along with the sensory analysis of the two time/temperature variations in the beer to determine the consumer awareness of TAB. The dependent variable in this study was the chemical breakdown of TAB. The chemical compounds were isolated and identified from the sampled beer resulting from the temperature abuse. The chemical breakdown was dependent on the two independent variables: storage time and temperature.

### 2.1. Beer Preparation

A Kölsch style bottled beer was used for this study. This type of beer was selected due to the assessment performed by Vanderhaegen et al. [18], who reported that lighter style beers represent the largest part of the beer market. The particular light style beer brand selected for this study was not pasteurized during the production process. The pasteurization process aids in the elimination of micro-organisms, which may cause spoilage in beer. Thus, the use of unpasteurized bottled beer may increase the likelihood of identifying volatile compounds.

As of December 2019, according to the Institute for Brewing Studies, there are now nearly 4522 microbreweries and 2594 brewpubs that produce 1,180,393 barrels of craft beer in Texas. In Texas, the number of craft breweries reached 341 in 2019 based on Brewers Association. Due to this growing segment in the beer industry, the researchers selected a local Texan craft microbrewery. The vast majority of craft beers are non-pasteurized [19]. The beer bottles were obtained directly off of the bottling line to avoid any prior time/temperature variations that might alter the data. (Table 1) presents the two manipulated time/temperature variations used in this study: Group A was the control (COB) group and was stored at the ideal conditions for storage at 3 °C for 35 days based on the Brewers Association [20]. Group A beer was stored in a Beverage Air Piedmont Line (PR24-1A, Spartenburg) refrigerator. This simulated group beer has been kept cold from the brewery to end consumer. Group B beer was stored at 35 °C for 30 days in an incubator (Fisher Scientific Isotemp Model 637D, Kansas City) and then stored at 3 °C for five days in a Beverage Air Piedmont Line refrigerator before sensory analysis. This transfer simulates retail or on-premise outlets that purchase the beer and then cool it before serving the consumer. The temperature of 35 °C simulates the storage temperature of non-refrigerated warehouses where beer is stored before being shipped to the various retail outlets for purchase by consumers and was chosen based on the results of Corzo and Brancho [21].

### 2.2. Chemical Analysis

After the two beer groups were exposed to temperature treatments for the specific time, they were packaged and shipped to Siebel Institute to conduct the chemical analysis in White Labs Analytical Services division in San Diego. Beer Groups A and B were packaged in an insulated container, packed with dry ice and expressed shipped overnight. The first comprehensive test (Complete QC Analysis) consisted of microbiological testing, alcohol content, extract values, calories, color, bittering units, pH, di-acetyl, and 2,3 pentanedione. The second complete test conducted was a flavor profile examination which focused on fusel alcohols and esters. To perform these chemical analyses, Siebel Institute utilizes two sets of scientific equipment (Siebel Institute of Technology [SIT], San Diego, CA, USA). The first set of scientific equipment that used was the Anton Paar Alcolyzer ME and Density Meter ME 5000 Haze QC ME Turbidity Meter were used to determine the alcohol content, extract values, calories, color, and the pH. The second set of scientific equipment that used for this testing was their Perkin Elmer Clarus 500 and TurboMatrix 110 Headspace Unit. This set of equipment identified the di-acetyl, 2,3 pentanedione, fusel alcohols, esters, bittering units, and sulphur dioxide content found in the two beer groups.

### 2.3. Participants for Sensory Analysis

After obtaining the appropriate approvals from the University of Houston, Internal Review Board (IRB), 147 participants were recruited from the student body, faculty, and staff. All participants were at least 21 and had no prior beer tasting training or education regarding the process of beer chemical breakdown. The decision was made to use untrained panelists due to the fact that untrained panels, when available, are advantageous in their more common availability, and the decreased time commitment resulting from the lack of training required [22]. In addition, the goal of the current study was to collect data from adults who would purchase and consume beer. For this study to achieve a desirable 95% confidence interval, the sample size needed was at least 147 participants conducting one trial each [22]. The experiment was designed for a significance level of α ≤ 0.05. Given the number of trials, and α ≤ 0.05, β ≤ 0.05 was determined with a Pd of 20% [22]. Therefore, the statistical power of this experiment was *p* ≥ 95% [21]. The critical number of correct responses in a triangle test with the above conditions was 60 participants in order for the results to be of significance [22].

### 2.4. Sensory Analysis

For this study, analytical discriminative sensory analysis was determined using a triangle test on the TAB at the Food Microbiology and Sensory Lab located in the University of Houston. A total of 147 participants completed one triangle test and were given random coded samples along with randomized variations in the two beer groups. Fifty milliliter samples were presented in Arton Viticole transparent 225 mL tasting glasses modeled after those found in ISO 3591 [23]. Each sample was marked with a randomly generated three-digit binding code along with a randomized variation of sample order that corresponded to those used by the SIMS 2000 software (SIMS Sensory Software Cloud), which is a computer software system for sensory evaluation, consumer insights, and market research data analyses.

The coded beer samples were then given to the assessors by passing them through the sliding stainless steel door with Group A as the COB Group. Participants were asked to sample each item from left to right to determine which sample was different from the other two. The triangle test used for this experiment was a forced-choice triangle test. Hence, if no noticeable differences were identified among the three samples, the program instructed the assessor to make a guess and select one sample to complete the test once concluded. The testers would select the appropriate binding code on their laptop, which gathered all panelist data anonymously, and stored it in a local SQL server for analysis. Testers were allowed as much time as possible to complete the test, however, no assessor took more than 10 min to complete it. All triangle tests performed for this study met the acceptable criteria found in norm ISO 4120 [24].

### 2.5. Data Analysis

For the sensory analysis, the independent variables were storage temperature and storage time, while the dependent variable was the noticeable sensory evaluation of the chemical breakdown of TAB. By utilizing beta analysis and a one-way ANOVA, it was possible to determine if there was a significant difference between the storage temperature and storage time (independent variables), and the detection through sensory evaluation of the chemical breakdown affecting the quality (dependent variable). The analyses also showed how the difference in beer temperature affected these results, thereby deciding whether the panel was able to determine a sensorial difference between the COB Group A beer and the treatment Group B beer. The SIMS 2000 software was used for data analysis. (SIMS Sensory Software Cloud, Berkeley Heights, NJ, USA).

## 3. Results and Discussion

### 3.1. Demographics

The average age of the 147 participants was 25 years. The oldest participant was 66 years old, and the youngest was 21. The occupations of the participants varied, but the vast majority were in the foodservice/hospitality industry. Of the 147 participants, 68 (46%) held foodservice/hospitality industry jobs. The second highest noted occupation was that of the classification of the student, which 47 (32%) of the 147 participants identified themselves as. The average job tenure of all of the respondents was 2 years. A total of 80 or (54%) had no managerial experience, while 67 (46%) stated that they have had at some point had managerial experience. A total of 111 (76%) were non-smokers, while 36 (25%) participants identified themselves as smokers. The average number of cigarettes smoked per day by smokers was nearly nine cigarettes per day. The average number of beers drank per week by all participants was a little more than eight. The data are demonstrated in Table 2.

### 3.2. Sensory Analysis

The sensory analysis of TAB yielded conclusive results based on the triangle test. A total of 147 participants conducted one forced-choice triangle test. Each participant was given three samples, yielding a total of 441 samples for the study (Table 3). A total of four sessions/day over one-week samples were tested. In each session, a total of 24 samples were completed. Minimum Number of Assessments in a Triangle Test corresponded to a Pd of 20% [22]. The triangle test conducted by the 147 participants yielded 57 correct responses when asked to select which beer sample is different from the other two. The 57 correct responses equate to 39% of participants being able to identify the correct beer. As shown in Table 3, the Critical Number of Correct Responses in a Triangle Test is 60. Therefore, 57 correct responses do not exceed the critical number of correct responses of 60 for the data to be significant [22]. Interpretation of the data states that 57 or more correct responses is evidence of a difference at the α = 0.05 level of significance. However, 56 or fewer correct responses indicates that a researcher can be 95% sure that no more than 20% of the participants can detect a difference, which is evidence of similarity relative to a P(d) = 20% at the β = 0.05 level of significance.

Two different analyses were utilized (beta analysis and one-way ANOVA analysis). Based on a beta analysis, the *p*-value was 0.0327 < the *p*-value of 0.05 level of significance. Thus, no significance was found in the beta analysis (Table 4). Analysis of a one-way ANOVA also shows no significance as the *p*-value was 0.0958 > the *p*-value of 0.05 level of significance (Table 4). Although the triangle tests yielded 90 incorrect responses, which equates to 60%, over one third of the participants could tell the difference between TAB and COB. Of the 147 participants, 91 (62%) were males. Of the 91 males, 32 or 22% produced correct responses. Of the 147 participants, 56 (38%) were female. Of the 56 females, 24 or 42% correct responses were produced. When the entire study population of 147 is taken into account the correct number of male responses of 32 (22%) and the female correct responses of 24 (16%). When the responses of 147 participants are taken into account, the incorrect number of male responses of 59 equates to 40% and the female incorrect responses of 31 equates to 21%.

The results from data analysis indicated that in a blind sensory analysis, consumers would not be able to detect sensorial flaws in beer caused by the chemical breakdown due to temperature abuse. The results from the current study are in agreement with those obtained by Mascia et al. [19]. The authors found differences among samples stored at a cold temperature (8 °C) and a high temperature (28 °C) in terms of chemical breakdown. However, the authors did not find significant changes in beer aroma from the results obtained by the sensory analysis, which conflicted with their hypothesis that the shelf life of different beers decreased due to high temperature and storage times. In the current study, although the critical number of correct responses did not show significance, the percentage of participants that could differentiate alterations among the samples was 39%. Previous research found that panelists who frequently consumed the sample beer were able to detect sensorial flaws in the beer with higher frequency than beers that they were not as familiar with or consumed less often [25]. In addition, if trained panelists were used, the critical number of correct responses would likely be met.

### 3.3. Chemical Analysis

The results of the chemical analysis conducted by the Siebel Institute of Technology are demonstrated in Table 5. The measurement of the apparent attenuation and real degree attenuation is slightly lower in the TAB by 0.03%, but it is not statistically different. Apparent attenuation is the percentage of the original extract that has been converted into CO2 and alcohol via the fermentation process. The lower levels show a drop in the level of fruity esters. This would suggest that the TAB had slightly lower sugar content than the COB. This could be due to some remaining live yeast cells in the beer converting the sugars into alcohol. Similar observation and explanation were also reported by Mascia et al. [19]. The alcohol by volume and the alcohol by weight are also slightly higher in the TAB than in the COB. This phenomenon could also be explained by some remaining live yeast in the beer samples. The remaining live yeast could have continued the fermentation process in the TAB due to the accelerated temperatures that the samples were exposed to. Future research of the same experimental processes should sample for live remaining yeast cells to confirm this hypothesis. The calorie count in the TAB is also slightly higher than the COB. This could be due to the increased alcohol content in the TAB, which could cause an increase in calories. The color of the TAB showed elevated levels on the Standard Reference Measurement (SRM) chart. This is a significant increase, due to the TAB having almost one point higher than the COB on the SRM color scale. This nearly one-point difference presented a noticeable color differentiation with the naked eye. This chemical reaction could be due to the Maillard reaction [26]. The Maillard reaction may be caused by several different heterocyclic compounds found in TAB [13,26] and could have been due to oxidation of the TAB [12]. The oxygen could have permeated through the crown cap of the bottle, although the Millard reaction does not require oxygen for the phenomenon to occur. The slight browning of the TAB could also have been caused by the complex reactions between the reduction in the sugars, proteins, peptides, and amino acids found in the beer samples [5]. Although there was no change in the real extract of the two beer groups, the apparent extract showed slightly higher levels in the TAB. This could be due to the alcohol content in the beer skewing the data of the hydrometer reading. Once the alcohol was removed, the real extract was the same in both beer samples. The increased levels of the original extract also indicate that there may be yeast cells present in the TAB. The pH of the TAB showed slightly elevated levels. This is also an indication of yeast cells still being present in the beer samples and is another indication that secondary fermentation continued in the TAB. The specific gravity of the two beer samples produced identical results in the chemical analysis data. This was expected because both beer samples were tested at 20 °C, which is one of the acceptable reference temperatures [27]. If the beer samples were tested at different temperatures, the specific gravity could have produced different results. This is because the density of the water in the beer varies with temperature. Thus, the testing of specific gravity at the three different reference temperatures (15.5, 17.5, and 20 °C) should all be done to obtain identical results and reliable data.

In the COB, there was no detection of acetaldehyde in the chemical analysis data. However, in the TAB, a considerably high level of acetaldehyde was detected. Acetaldehyde typically gives the beer a characteristic of apples or a crisp, refreshing sensation [28]. The level detected in the TAB was 3.74 mg/L. This suggests that the TAB was exposed to oxygen. This chemical formation suggests that oxygen somehow forced its way through the crown cap of the bottle. The bottle of the TAB in the incubator remained the same size, while at the same time, the crown cap may have expanded, allowing for oxygen to enter the bottle. Although in some beers the presence of acetaldehyde gives the beer its desired flavor profile, this was not the case with this particular beer sampled. Another ester that showed decreased levels in the TAB was ethyl acetate. The decrease in this chemical compound was, at −10.04 mg/L, less than the COB. Neven et al. [29] suggested that this decrease may be caused by extracellular esterases released from the yeast that may catalyze the breakdown of the esters. Combining this phenomenon with the chemical hydrolysis of esters could have produced this significant decrease in ethyl acetate. The levels of both isoamyl acetate and isoamyl alcohol showed a decrease in the TAB. Isoamyl acetate is produced by the esterification of isoamyl alcohol. The decrease in both suggests that there were still live yeast cells in the samples. Our results are similar to results obtained by Vanderhaegen et al. [18], who reported that the process of hydrolysis decreased the concentrations of the esters. There was no detection of acetone in either beer group due to acetone being a major flaw in beer that creates a solvent type flavor. The time of the TAB was perhaps not long enough to develop this chemical compound. Usually, acetone is developed due to prolonged exposure to high temperatures during the fermentation process [30]. There was an increase in ethyl octanoate in the TAB. Ethyl octanoate produces a cognac-apricot taste in beer. No ethyl hexanoate was detected in the COB, and the TAB showed detectable levels of 0.30 mg/L. This increase raised it over the limit of the human threshold. This increase imparted more of an anise flavor and aroma that the COB did not have. The increase in both of these ethyl esters is likely due to hydrolysis during the forced aging temperature of 35 °C. Although the previous two ethyl esters showed increased levels in the TAB, the chemical compound ethyl butyrate showed decreased levels in the TAB. However, the detected levels were below the threshold of 0.4 mg/L. As expected, there was no detection of methanol in either beer group. In both congener alcohols tested, there was a decrease in the levels of the TAB samples. The decrease in the levels of isobutanol in the TAB was minimal. However, the decrease in 1-proponal was considerably different. The SO_2_ chemical analysis showed increase levels in the TAB of +0.39 mg/L. The threshold level is one mg/L, so an increase of 0.39 mg/L is significant when the SO_2_ levels of the TAB are 2.84 mg/L. This elevated level made the TAB considerably more bitter than the COB. With the increased levels of SO_2_, the levels of IBUs in the TAB showed an increase of +0.9 BU. This increase is almost an entire unit on the BU scale. The nearly one-unit increase in the TAB could be attributed to the decrease in the esters that impart a fruitful flavor in the beer. The higher level of BU in the TAB could be due to the presence of a higher level of polyphenol in the TAB. The increase in polyphenols in beer is responsible for bitterness [30]. Additionally, flavonoids, which are one of the polyphenols, are known to cause permanent beer haze during storage. However, despite the negative constitution with beer haze formation, polyphenols act as antioxidants, preventing oxidative degradation of beer [30]. Both of the vicinal diketones that were analyzed showed increased levels in the temperature beer. Diacetyl (2,3-butanedione) and 2,3-pentanedione are two important chemical components that significantly affect beer flavor. Diacetyl is known for having an extensive and serious off-flavor component in beer and has a potent butterscotch aroma at concentrations above the flavor threshold, ranging from 0.1 to 0.15 ppm in lager beers. Diacetyl is reduced by yeast to acetoin and 2,3-butanediol, compounds with relatively high flavor thresholds at the end of the main fermentation and maturation phase [31]. Diacetyl and 2, 3 pentanedione showed elevated levels due to the Maillard reaction. The reaction was due to the high storage temperature of the beer [31].

The chemical analysis yielded concise results that support our hypothesis, but conflicted with the sensory analysis results. In the sensory analysis, the number of correct responses were 57, and it was very close to the critical number of 60. However, the critical number of correct responses did not meet significance. This may be because untrained panels were used for the sensory analyses, and most of the volatile components were below the perception threshold. Therefore, trained panels able to identify the differences in flavor between different groups of beer samples better than untrained panels. To this end, the sensory analyses conflicted with chemical analysis results. Although the sensory analysis results were not at the level of significance to show participant identification, the data results obtained from the chemical analyses showed definite chemical changes in the two beer groups.

## 4. Conclusions

Flavor instability due to beer storage and temperature remains one of the most critical quality problems in the brewing industry. The results of this study have provided the beer industry with viable implications by reconfirming the principles on how to properly store beer and avoid temperature fluctuations in order to maintain a quality product. This study has proven that TAB has chemical alterations that affect the quality of the product, that beer producers need to take into consideration. These alterations cause overall differences in taste and quality that can affect consumer perception in a negative manner, deterring repeated sale of the product.

The study used a convenience sampling of untrained assessors to best represent the average beer drinker. Future research may benefit from an expert trained panel for the sensory analysis. In addition, some non-pasteurized craft beers that are not as popular as others may endure temperature abuse much longer than one month. Furthermore, only one type of craft non-pasteurized beer was used in this experiment. Other types of beer may produce different chemical changes in higher frequencies than the beer type selected for this experiment. The light and vibration abuse that may affect the chemical and sensorial properties of the beer were excluded from this experiment. Future research may include simulations of light abuse and vibration abuse that beer products may encounter during the shipment process.

## Figures and Tables

**Table 1 foods-09-01032-t001:** Beer Group Time/Temperature Treatment *.

Beer Group	Temperature	Time (Days)
A	3 °C (±0.5 °C)	35
B	35 °C (±0.2 °C)	30
B	3 °C (±0.5 °C)	5

* All beer served at 3 °C during sensory analysis.

**Table 2 foods-09-01032-t002:** Demographic Response Totals.

Gender	Males = 91 Females = 56
**Average Age**	24.8 Years Old
**Occupation**	Food and Beverage Service = 59 Education = 51 Other Occupation = 17 Sales = 11 Hotel = 9
**Average Length of Job Tenure**	1.9 Years
**Managerial Position**	No = 80 Yes = 67
**Smoker**	No = 111 Yes = 36
**Average # of Cigarettes Smoked Per Day**	9.7 Cigarettes Per Day
**Average # of Beers Consumed Per Week**	8.3 Beers Per Week

**Table 3 foods-09-01032-t003:** Sensory analysis inputs and outputs.

Inputs	Outputs
Number of Respondents: *n* = 147	Probability of a Correct Response @ p(d): Pc = 0.4666
Number of Correct Responses: x = 57	TYPE I Error: α-risk = 0.0500
Probability of a Correct Guess: Pc = 0.333	TYPE II Error: β-risk = 0.0500
Proportion Distinguishers: P(d) = 0.20	Power: 1-β = 0.9500

**Table 4 foods-09-01032-t004:** Beta analysis results and one-way ANOVA results for sensory evaluation.

**Beta Analysis**								
Attribute	Probability (Pc)	Trials (N)	Successes (M)	Absent	Pd	Beta *p*-Value	Sig When Null Hyp = No Diff	*^a^* Sig When Null Hyp = Diff
Triangle	0.46667	147.0	57.0	0.0	0.20	0.0327	N/A	*^b^* NS
**ANOVA Results**								
Attribute	Probability (Pc)	Trials (N)	Successes (M)	One Tailed *p*-Value			*^a^* Sig	
Triangle	0.33333	147.0	57.0	0.0958			*^b^* NS	

*^a^* Sig refers to Significant; *^b^* NS refers to not significant; Hyp refers to hypothesis; Diff refers to different; *n* = number of participants, M = number of correct responses; N/A = not applicable; Pd = minimum number of assessors; Pc probability of correct responses.

**Table 5 foods-09-01032-t005:** Chemical Analysis Results.

Test	Method	Group A (COB) Mean	Group B (TAB) Mean	Difference of Beer Groups	Measurement in Units
Apparent Attenuation	ASBC Beer-6C	87.12	87.09	−0.03	Percent
Alcohol by Volume	ASBC Beer-4C	5.23	5.24	+0.01	Percent
Alcohol by Weight	ASBC Beer-4C	4.11	4.12	+0.01	Percent
Calories	ASBC Beer-33	147.76	148.04	+0.28	*^a^* Per 12 fl. Oz.
Color	ASBC Beer-10A	7.08	7.81	+0.73	SRM
Apparent Extract	ASBC Beer-3	1.46	1.47	+0.01	Plato
Real Extract	ASBC Beer-5A	3.37	3.37	Same	Plato
Original Extract	ASBC Beer-6A	11.36	11.38	+0.02	Plato
pH	ASBC Beer-9	4.30	4.32	+0.02	
Real Degree Attenuation	ASBC Beer-6B	71.62	71.60	−0.02	Percent
Specific Gravity	ASBC Beer-2B	1.006	1.006	Same	20 C
Acetaldehyde	ASBC Beer-29 M	ND	3.74	+3.74	mg/L
Isoamyl Acetate	ASBC Beer-29 M	1.01	0.32	−0.69	mg/L
Ethyl Octanoate	ASBC Beer-29 M	0.20	0.34	+0.14	mg/L
Ethyl Hexanoate	ASBC Beer-29 M	ND	0.30	+0.30	mg/L
Acetone	ASBC Beer-29 M	ND	ND	Same	mg/L
Ethyl Butyrate	ASBC Beer-29 M	0.33	0.16	−0.17	mg/L
Methanol	ASBC Beer-29 M	ND	ND	Same	mg/L
Isoamyl Alcohol	ASBC Beer-29 M	44.33	39.09	−5.24	mg/L
Isobutanol	ASBC Beer-29 M	14.22	9.89	−4.33	mg/L
Ethyl Acetate	ASBC Beer-29 M	30.36	20.32	−10.04	mg/L
1-Propanol	ASBC Beer-29 M	32.69	5.96	−26.73	mg/L
SO_2_	ASBC Beer-21A	2.45	2.84	+0.39	mg/L
Diacetyl	ASBC Beer-25E	46.93	49.58	+0.00265	mg/L
2,3 Pentanedione	ASBC Beer-25E	ND	3.95	+0.00395	Mg/L
Bitterness Units	ASBC Beer-23A	15.0	15.9	+0.9	BU

All analyses were carried out in duplicate; COB = Control beer; TAB = temperature abused beer. *^a^* fl. Oz (fluid ounce) is a unit of volume typically used for measuring liquids, 1 US fl Oz = 0.03 L. ND = none detected.

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
