# Peer review of "Implications of Temperature Abuse on Unpasteurized Beer Quality Using Organoleptic and Chemical Analyses"

_foods, 2020, doi:10.3390/foods9081032_

Round 1
Reviewer 1 Report
Manuscript Number: foods-867022
Title: Implications of temperature abuse on unpasteurized beer quality using organoleptic and chemical analyses
The manuscript reports the study of the abuse of temperature during beer storage. This is an important aspect of the beer quality because it is well known that the temperature deeply affect the sensory quality of beer. However the results of the manuscript do not fit with these statements. The authors should justify better their results. The bibliography must be improved because some works about the sensory aspect and the effect of temperature are not cited (ie. Journal of the Science of Food and Agriculture, 94 (2014) 919–928; Journal of the Science of Food and Agriculture, 96, (2016), 4106-4115). Moreover some references in the text are not correct or do not correspond to the reference list (see line 223 Mascia et al.).
In particular:
Lines 235-237: the referee suggests to report a more appropriate reference. The references 12, 13, 25 are not about Maillard reaction.
Line 215: Chemical analysis section. The referee suggests to revise the discussion of these results after an appropriate statistical analysis of the data reported in table 6. In fact the authors must report the analytical replicas, the standard deviation and the statistical differences between the two groups of beers stored at different temperature. The statistical analysis is important because is very difficult affirm that the values of apparent attenuation, alcohol calories, extract, pH are different without to consider the standard deviation or the uncertainty of the methods.
Line 297: can you explain better this affirmation? Can you reported an appropriate reference? Generally the vicinal diketones are due to the yeast metabolism.
Lines 299-302: the authors must conclude the work taking into account the statistical analysis of the data (see above) and the fact that all the volatile compounds, except ethyl hexanoate, are below the perception threshold. It is very difficult for untrained participants to the sensory analysis perceive difference. Revise the conclusion please.

Author Response
Response to Reviewer 1 Comments
Point 1: The manuscript reports the study of the abuse of temperature during beer storage. This is an important aspect of the beer quality because it is well known that the temperature deeply affect the sensory quality of beer. However, the results of the manuscript do not fit with these statements. The authors should justify better their results. The bibliography must be improved because some works about the sensory aspect and the effect of temperature are not cited (ie. Journal of the Science of Food and Agriculture, 94 (2014) 919–928; Journal of the Science of Food and Agriculture, 96, (2016), 4106-4115). Moreover, some references in the text are not correct or do not correspond to the reference list (see line 223 Mascia et al.).
Response 1: Thank you for comment. We have corrected the order of the references and have cited the recommended references and added to our references list. The following explanation and the reference was added to the introduction section.
“Marconi et al. [14] conducted two different experimental conditions where beer samples were either stored at 20 °C or 30 °C to investigate the influence of yeast strain, priming solution, and temperature on beer quality. The researchers also investigated the standard quality attributes, volatile compounds, and sensory profile of the bottle-conditioned beer. The authors found that the volatile profile was affected by the strain of yeast due to different metabolic activity for each yeast strain. The authors also found that the Safbrew S-33® yeast strain, when primed with Siromix® and refermented at 30 °C, yielded the fastest formation of higher alcohols while maintaining low production of off-flavors. Finally, the authors concluded that a formulation of two yeast strains might reduce the time needed for bottle conditioning without affecting the quality of the final beer and improve efficiency and economic profits [14].”
Marconi, O., Rossi, S., Galgano, F., Sileoni, V., & Perretti, G. (2016). Influence of yeast strain, priming solution and temperature on beer bottle conditioning. Journal of the Science of Food and Agriculture, 96(12), 4106-4115.
Point 2: Lines 235-237: the referee suggests to report a more appropriate reference. The references 12, 13, 25 are not about Maillard reaction.
Response 2: We appreciate your comment. The references order numbers have been corrected. Please see the number of references that talk about Maillard reaction and cited text.
Point 3: Line 215: Chemical analysis section. The referee suggests to revise the discussion of these results after an appropriate statistical analysis of the data reported in table 6. In fact, the authors must report the analytical replicas, the standard deviation and the statistical differences between the two groups of beers stored at different temperature. The statistical analysis is important because is very difficult affirm that the values of apparent attenuation, alcohol calories, extract, pH are different without to consider the standard deviation or the uncertainty of the methods.
Response 3: We appreciate your comment. The chemical analyses in Table 6 was performed by the Siebel Institute of Technology and the data are the means of the analyses. Additionally, we received the analyses in its current form as means without the standard deviation. We have added the methods to the Table 6.
Point 4: Line 297: can you explain better this affirmation? Can you report an appropriate reference? Generally, the vicinal diketones are due to the yeast metabolism.
Response 4: Thank you for the comment. Additional explanation regarding vicinal diketones was added to the manuscript. We have cited the following reference as a source of this information.
Olaniran, A. O.; Hiralal, L.; Mokoena, M. P.; Pillay, B. Flavour‐active volatile compounds in beer: production, regulation and control. J. Inst. Brew. 2017, 123,13-23.
https://doi.org/10.1002/jib.389
Point 5: Lines 299-302: the authors must conclude the work taking into account the statistical analysis of the data (see above) and the fact that all the volatile compounds, except ethyl hexanoate, are below the perception threshold. It is very difficult for untrained participants to the sensory analysis perceive difference. Revise the conclusion please.
Response 5: We appreciate your comment. We have made changes in our conclusion section based on your suggestion.

Reviewer 2 Report
In their Manuscript ID: foods-867022 entitled "Implications of temperature abuse on unpasteurized beer quality using organoleptic and chemical analyses" for Foods, authors have reported somewhat original data and I think that the study is very interesting and it deserves to be published.
However, some points of criticism and questions have to be clarified prior publication:
The most important point is that reference number and listed references did not match! For example: references in lines 37, 71, 93, 101, 136, 150, 267, 273… And they were the clearly identifiable, but the majority of the references are wrong. This is a very important aspect that MUST BE CORRECTED. The good discussion is really damaged because this sadly lack. ALL REFERENCES must be revised, verified and, if necessary, corrected in all the sections of the document.
ABSTRACT:
- Line 19-20: The sentence “These results can […] the distribution chain” should be the last sentence of the Abstract.
- Lines 20-22: In my opinion, this is not a general conclusion of the present work, but a particular result. I think a global conclusion should be included in the abstract.
INTRODUCTION:
- LINE 47: I think that these compounds should be better not starting by capital letters.
- LINE 55: Remove “30 ppb” to maintain the homogeneity of the text.
- LINE 6: Table 1 should be removed because it presents data not analyzed. Instead, the chemical class can be indicated in brackets in Table 6 for each chemical compound in the first column.
- LINES 75-78: I think the letters have different form or size. It should be corrected.
- LINE 86: Where? In USA? In the World? The geographical extension of this information should be given.
- LINE 91: Not demonstrates, but presents.
- Table 2 is not clear. I would remove the line A&B. I would indicate directly that Group A has passed 35 days at 3°C. I would indicate somehow that Group B has passed 30 days at 35°C and later 5 days at 3°C. Moreover, in the text, is not written that Group A has passed 35 days at 3°C (and not 30). This last point must be corrected.
- LINE 112: Not only “Siebel”, but “Siebel Institute”.
- LINE 114: (Siebel Institute of Technology –SIT– should be moved to line 113 following to equipment.
- 113 Turbidity Meter? This point should be verified.
- LINE 116: The Methods to measure the different parameters are not described. This is the second most important point. The Methods for each parameter MUST BE DESCRIBED.
- LINES 113 and 116: Data about the company/country that produced that equipment?
- LINE 129: p in cursive
- LINE 142: group in capital letter (Group).
- LINE 160: Data about the company/country that produced that software?
RESULTS AND DISCUSSION
- LINES 162 to 174: This is not Results or Discussion. This paragraph should be moved to line 131 (Section 2.3).
- LINES 163-164. Race distribution should be removed. In my opinion, it is not relevant. I think that it would be better to collect some cultural aspect such as “time living in USA” which would better reflect the cultural preferences because of living country. Similarly, Race distribution should be removed of Table 3. Fromm my personal point of view, this point could imply ethical concerns.
- LINE 165: 25 years (to facilitate the reading)
- LINE 169: nearly 2 years (to facilitate the reading)
- LINE 172 and 173: “nearly 9 cigarettes” and “a little more than 8” (to facilitate the reading).
- LINE 174: Table 3 not in brackets.
- LINE 179: Did participants taste 441 samples the same day? The same week? The same month? I think it is important to contextualize the number of samples tasted in each tasting session and the total duration of the sensory analysis.
- LINES 182, 194, 198, 199, 200: 39%, 61%, 22%, 22%, 16%, 40% (to facilitate the reading)
- LINE 196: Calculate and include in the text the % of males (91 of 147).
- LINE 197: Calculate and include in the text the % of females (65 of 147).
- Table 5: All abbreviations must be defined (Sig, Hyp, Diff, N/A, NS…). Moreover, (N) and (M) must be defined. Finally, the columns should be bigger to avoid “Attribut” in one level and “e” in other level (similar in the case of Probabilit and y, Trial and s, SUccesse and s, Absen and t…).
- LINE 208: Did these authors dive some hypothesis? If affirmative, it would be interesting to include it.
- LINE 214: It could have been very interesting and valuable to do the experiment also with trained panelists.
- LINE 218: 0.03%...Is this statistically different?
- LINE 217: Table 6: not in brackets.
- LINES 222 and LINES 245. In my opinion, it seems very contradictory that authors suggested that live yeasts cells remains in control beer (LINE 222 and 225), but secondary fermentation (LINE 245 and 247) appeared in temperature abused beer. This point MUST BE DISCUSSED and clarified.
- IN ALL THE TEXT: Cannot be “temperature abused beer” abbreviated as TBA and control beer abbreviated as COB? I think it would facilitate the reading.
- Table 6: Calories: Per 12 fl. Oz…define fl. and Oz (not all readers are familiarized with this).
- LINE 286: Calculate and change 400 ppb to mg/L, to maintain the homogeneity of the text.
- LINE 299: A little summary about the conflicts between chemical analysis and sensory analysis results would be appreciated.
There is no particular major point to review (just important points to review, such as the wrongly match between references in the text and listed references, and the no-described methodology). The number of points has varying level or importance; I suggest MINOR REVISION to the Editor.
ADDITIONAL CONCERNS:
Do you think that the analysis of polyphenols (related to LINE 50) could be of interest to research? And the foam/foamability, keeping in mind that it is a major parameter in beer? It would be also interesting include not only differentiation by tasting participants but also their preferences. It would be also interesting to do the sensorial analysis with “black glasses”, in order to not be influenced by color and only focus in flavor parameters.
In my opinion, these points could be convenient to research in next times (not in the present work, just suggestion for further studies) and would enrich your valuable work. Congratulations to the authors for this interesting study.

Author Response
Response to Reviewer 2 Comments
Authors have reported somewhat original data and I think that the study is very interesting and it deserves to be published.
However, some points of criticism and questions have to be clarified prior publication:
Point 1: The most important point is that reference number and listed references did not match! For example: references in lines 37, 71, 93, 101, 136, 150, 267, 273… And they were the clearly identifiable, but the majority of the references are wrong. This is a very important aspect that MUST BE CORRECTED. The good discussion is really damaged because this sadly lack. ALL REFERENCES must be revised, verified and, if necessary, corrected in all the sections of the document.
Response 1: We appreciate the comment. The order of all references have been corrected in the manuscript and number represent the actual references.
ABSTRACT:
Point 2: Line 19-20: The sentence “These results can […] the distribution chain” should be the last sentence of the Abstract.
Response 2: Thank you for the comment. The sentence in Line 19-20 that starts with “These results” was moved to the last sentence in the abstract.
Point 3: Lines 20-22: In my opinion, this is not a general conclusion of the present work, but a particular result. I think a global conclusion should be included in the abstract.
Response 3: Thank you for the comment. A global conclusion sentence has been added to the abstract.
INTRODUCTION:
Point 4: LINE 47: I think that these compounds should be better not starting by capital letters.
Response 4: Thank you for the comment. The chemical compounds were changed in the manuscript text.
Point 5: LINE 55: Remove “30 ppb” to maintain the homogeneity of the text.
Response 5: Thank you for the suggestion. We have corrected the manuscript based on your recommendation.
Point 6: LINE 6: Table 1 should be removed because it presents data not analyzed. Instead, the chemical class can be indicated in brackets in Table 6 for each chemical compound in the first column.
Response 6: Thank you for the comment. We have removed Table 1 from the manuscript. We have not added chemical class to Table 6 because these classifications are mentioned in the text.
Point 7: LINES 75-78: I think the letters have different form or size. It should be corrected.
Response 7: We appreciate the comment. The font was corrected in the manuscript text.
Point 8: LINE 86: Where? In USA? In the World? The geographical extension of this information should be given.
Response 8: Thank you for the comment. The information was in the State of Texas. This has been added to the manuscript text.
Point 9: LINE 91: Not demonstrates, but presents.
Response 9: We appreciate your suggestion. The word demonstrates was changed to presents in the manuscript.
Point 10: Table 2 is not clear. I would remove the line A&B. I would indicate directly that Group A has passed 35 days at 3°C. I would indicate somehow that Group B has passed 30 days at 35°C and later 5 days at 3°C. Moreover, in the text, is not written that Group A has passed 35 days at 3°C (and not 30). This last point must be corrected.
Response 10: We appreciate this comment. A&B was removed in Table 2. We added B as stored at 3°C for 5 days and changed 30 days for Group A in Table 2 and manuscript text
Point 11: LINE 112: Not only “Siebel”, but “Siebel Institute”.
Response 11: Thank you for the comment. The word Institute was added to the manuscript.
Point 12: LINE 114: (Siebel Institute of Technology –SIT– should be moved to line 113 following to equipment.
Response 12: We appreciate your suggestion. Based on your recommendation, (Siebel Institute of Technology [SIT], San Diego) was moved in the manuscript.
Point 13: 113 Turbidity Meter? This point should be verified.
Response 13: We appreciate your comment. Turbidity meter was part of the first scientific set of equipment that Siebel Institute used for chemical analysis. This was included in the manuscript.
Point 14: LINE 116: The Methods to measure the different parameters are not described. This is the second most important point. The Methods for each parameter MUST BE DESCRIBED.
Response 14: We appreciate your comment. The chemical analyses were performed by the Siebel Institute of Technology. We have included a new column for the methods for each chemical component in Table 6.
Point 15: LINES 113 and 116: Data about the company/country that produced that equipment?
Response 15: Thank your comments. The equipment was used for the chemical analyses by Siebel Institute.
Point 16: LINE 129: p in cursive
Response 16: Thank you the comment. p was italicized and capitalized.
Point 17: LINE 142: group in capital letter (Group).
Response 17: Thank you for the comment, the group was capitalized for both group A and control group in the manuscript.
Point 18: LINE 160: Data about the company/country that produced that software?
Response 18: Thank you for the suggestion, the information about the company and country was added to the manuscript.
RESULTS AND DISCUSSION
Point 19: LINES 162 to 174: This is not Results or Discussion. This paragraph should be moved to line 131 (Section 2.3).
Response 19: We appreciate your comment. we have included the demographic details in the results and discussion section. We included total of participant and the location in the materials and methods section.
Point 20: LINES 163-164. Race distribution should be removed. In my opinion, it is not relevant. I think that it would be better to collect some cultural aspect such as “time living in USA” which would better reflect the cultural preferences because of living country. Similarly, Race distribution should be removed of Table 3. Fromm my personal point of view, this point could imply ethical concerns.
Response 20: Thank you for the comment. We removed the section related to the race from demographic part in both the text and in Table 3
Point 21: LINE 165: 25 years (to facilitate the reading).
Response 21: Thank you the suggestion, this was changed in the manuscript.
Point 22: LINE 169: nearly 2 years (to facilitate the reading).
Response 22: Thank you for the comment, this was changed in the manuscript.
Point 23: LINE 172 and 173: “nearly 9 cigarettes” and “a little more than 8” (to facilitate the reading).
Response 23: Thank you for the comment, both numbers were changed in the manuscript.
Point 24: LINE 174: Table 3 not in brackets.
Response 24: Thank you for the comment, the brackets were removed from the Table 3 in line 174.
Point 25: LINE 179: Did participants taste 441 samples the same day? The same week? The same month? I think it is important to contextualize the number of samples tasted in each tasting session and the total duration of the sensory analysis.
Response 25: Thank you for the suggestion, the following explanation was added to text “A total of 4 sessions/day over one-week samples were tested. In each session, a total of 24 samples were completed”.
Point 26: LINES 182, 194, 198, 199, 200: 39%, 61%, 22%, 22%, 16%, 40% (to facilitate the reading).
Response 26: Thank you for the comment, the percentages were rounded to be 39%, 22%, 22%, 16%, 40% in the manuscript.
Point 27: LINE 196: Calculate and include in the text the % of males (91 of 147).
Response 27: Thank you for the comment, the % of 91 of 147 males was calculated and added to the manuscript.
Point 28: LINE 197: Calculate and include in the text the % of females (56 of 147).
Response 28: Thank you for the comment, the % of 56 of 147 female was calculated and added to the manuscript.
Point 29: Table 5: All abbreviations must be defined (Sig, Hyp, Diff, N/A, NS…). Moreover, (N) and (M) must be defined. Finally, the columns should be bigger to avoid “Attribut” in one level and “e” in other level (similar in the case of Probabilit and y, Trial and s, SUccesse and s, Absen and t…).
Response 29: Thank you for the comment, Table 5 was corrected, and all abbreviations were defined as a note in Table 6 of the manuscript.
Point 30: LINE 208: Did these authors dive some hypothesis? If affirmative, it would be interesting to include it.
Response 30: Thank you for the comment. We have added the authors' hypothesis to line 208, which is line 230-231 in the revised manuscript.
Point 31: LINE 214: It could have been very interesting and valuable to do the experiment also with trained panelists.
Response 31: Thank you for your comment. We agree with your suggestion; however, due to the COVID-19 pandemic we will not be able to conduct this follow up research in a timely manner. This would be our priority for our future research project.
Point 32: LINE 218: 0.03%...Is this statistically different?
Response 32: Thank you for the comment, this is not statistically different. This information was added to the manuscript.
Point 33: LINE 217: Table 6: not in brackets.
Response 33: Thank you for the suggestion, the brackets were removed from Table 6 in line 217.
Point 34: LINES 222 and LINES 245. In my opinion, it seems very contradictory that authors suggested that live yeasts cells remains in control beer (LINE 222 and 225), but secondary fermentation (LINE 245 and 247) appeared in temperature abused beer. This point MUST BE DISCUSSED and clarified.
Response 34: Thank you for the comment. This was corrected in the manuscript.
Point 35: IN ALL THE TEXT: Cannot be “temperature abused beer” abbreviated as TAB and control beer abbreviated as COB? I think it would facilitate the reading.
Response 35: Thank you for the suggestion. The abbreviation for temperature abused beer (TAB) and control beer (COB) was added to the manuscript.
Point 36: Table 6: Calories: Per 12 fl. Oz…define fl. and Oz (not all readers are familiarized with this).
Response 36: Thank you for the comment, the fl. Oz. was defined and added to the manuscript as note of Table 6.
Point 37: LINE 286: Calculate and change 400 ppb to mg/L, to maintain the homogeneity of the text.
Response 37: Appreciate your suggestion, 400 ppb was calculated to be 0.4 mg/L in the manuscript.
Point 38: LINE 299: A little summary about the conflicts between chemical analysis and sensory analysis results would be appreciated.
Response 38: We appreciate your comment, we have added a summary about the conflict between chemical and sensory analysis in the manuscript.
There is no particular major point to review (just important points to review, such as the wrongly match between references in the text and listed references, and the no-described methodology). The number of points has varying level or importance; I suggest MINOR REVISION to the Editor.
ADDITIONAL CONCERNS:
Do you think that the analysis of polyphenols (related to LINE 50) could be of interest to research? And the foam/foamability, keeping in mind that it is a major parameter in beer? It would be also interesting include not only differentiation by tasting participants but also their preferences. It would be also interesting to do the sensorial analysis with “black glasses”, in order to not be influenced by color and only focus in flavor parameters.
In my opinion, these points could be convenient to research in next times (not in the present work, just suggestion for further studies) and would enrich your valuable work.
Response: Thank you for sharing your insight. We will keep these in mind for future research.
Reviewer 3 Report
The authors examined the changes in unpasteurized beer exposed to different temperatures. The research is very interesting and can be utilized in the industry, especially since many of the breweries employ packaging in cans or bottles. My remarks:
- lines 75-78 - please correct the font.
- line 86-90 - this part does not belong to M&M. This could be transferred to the Introduction.
- in the introduction, you are mentioning the influence of polyphenols on oxidation and staling of beers. I would like to see you include this part into the discussion and I would like to recommend including the analysis of polyphenols for future research.
Author Response
Response to Reviewer 3 Comments
The authors examined the changes in unpasteurized beer exposed to different temperatures. The research is very interesting and can be utilized in the industry, especially since many of the breweries employ packaging in cans or bottles. My remarks:
Point 1: lines 75-78 - please correct the font.
Response 1: The font was corrected in the manuscript text.
Point 2: line 86-90 - this part does not belong to M&M. This could be transferred to the Introduction.
Response 2: We appreciate your comment, we have included this section to clarify why we have chosen the Texas non-pasteurized craft beers, and we think, it would be valuable for the reader to have this section in materials and methods. However, we have added the phrase to the introduction that related to this section.
Point 3: in the introduction, you are mentioning the influence of polyphenols on oxidation and staling of beers. I would like to see you include this part into the discussion and I would like to recommend including the analysis of polyphenols for future research.
Response 3: Thank you for your comment, we will include the analysis of polyphenols in our future research. We have added this information about the influence of polyphenols on beer quality to discussion section “The higher level of BU in the TAB could be due to the presence of a higher level of polyphenol in the TAB. The increase in polyphenols in beer is responsible for bitterness. Additionally, flavonoids, which is one of the polyphenols, are known to cause permanent beer haze during storage. However, despite the negative constitution with beer haze formation, polyphenols act as antioxidants, preventing oxidative degradation of beer”. This information has been added to the last paragraph in discussion section. We have also added the following reference and cited in the text.
Oladokun, O.; Tarrega, A.; James, S.; Smart, K.; Hort, J.; Cook, D. The impact of hop bitter acid and polyphenol profiles on the perceived bitterness of beer. Food Chem. 2016, 205, 212-220. https://doi.org/10.1016/j.foodchem.2016.03.023
Round 2
Reviewer 1 Report
This referee suggest again to report standard deviation and statistical analysis in table 6.
Author Response
Point 1: This referee suggest again to report standard deviation and statistical analysis in table 6.
Response 1: Thank you for your comment. The chemical analyses component of this study was performed by White Labs and we contacted them to request the raw data. However, this study was performed a few years ago and unfortunately, they do not have the original data anymore. Nevertheless, we still have some good points to share. First, based on the lab data, the multiple analyses conducted by the white lab and the variation between the data were too small, and that was one reason that the standard deviations were not recorded for this data. Second, we have added some information to table 6, including the methods, fixing the caption to reflect the means, which is the actual means.